# Integrated Transcriptome and Metabolome Analysis Identifies Key Genes Regulating Maize Tolerance to Alkaline Stress

**DOI:** 10.3390/ijms262110632

**Published:** 2025-10-31

**Authors:** Shouxu Liu, Zichang Jia, Xuanxuan Hou, Xue Yang, Fazhan Qiu, Meisam Zargar, Moxian Chen, Congming Lu, Yinggao Liu

**Affiliations:** 1National Key Laboratory of Wheat Improvement, College of Life Sciences, Shandong Agricultural University, Tai’an 271018, China; liushou15963@163.com (S.L.); 18854806295@163.com (X.H.); 2State Key Laboratory of Green Pesticide, Key Laboratory of Green Pesticide and Agricultural Bioengineering, Ministry of Education, Center for R&D of Fine Chemicals of Guizhou University, Guiyang 550025, China; jiazc973@163.com (Z.J.); xueyang202001@163.com (X.Y.); 3National Key Laboratory of Crop Genetic Improvement, Huazhong Agricultural University, Wuhan 430070, China; qiufazhan@mail.hzau.edu.cn; 4Department of Agrobiotechnology, Institute of Agriculture, Peoples’ Friendship University of Russia, Moscow 117198, Russia; zargar_m@rudn.ru; 5National Key Laboratory for the Development and Utilization of Forest Food Resources, State Key Laboratory of Tree Genetics and Breeding, Key Laboratory of State Forestry and Grassland Administration on Subtropical Forest Biodiversity Conservation, The Southern Modern Forestry Collaborative Innovation Center, College of Life Sciences, Nanjing Forestry University, Nanjing 210037, China; cmx2009920734@gmail.com

**Keywords:** alkaline stress, maize, metabolome, organic acids, root system, transcriptome

## Abstract

Soil salinization threatens global food security, necessitating the development of saline–alkaline-tolerant crops. This study investigated the molecular mechanisms of alkali stress tolerance in maize. Screening 369 inbred lines identified two alkali-resistant and two alkali-sensitive varieties. Systematic analysis revealed that resistant varieties rapidly lowered rhizosphere pH and maintained root architecture, whereas sensitive varieties suffered reduced lateral roots and severe biomass loss. Metabolomic profiling showed that all varieties secreted malonic acid via the pyrimidine pathway to modulate rhizosphere pH, with resistant varieties exhibiting stronger accumulation. Transcriptome and RT-qPCR analysis identified two key genes: *Zm00001eb396990* (asparagine synthetase), upregulated in resistant varieties and linked to organic acid synthesis, and *Zm00001eb370000* (cytokinin dehydrogenase), downregulated in resistant varieties, potentially aiding root maintenance. Multi-omics correlation confirmed the association between *Zm00001eb396990* expression and malonic acid content. This study demonstrates that maize roots can alleviate alkali stress through the secretion of malonic acid and the regulation of related genes, providing potential genetic targets and a theoretical basis for cultivating alkali-tolerant maize.

## 1. Introduction

Soil alkalization has emerged as a critical factor imperiling global food security [1,2,3]. According to partial statistics from the Food and Agriculture Organization of the United Nations (FAO) in 2021 [4], globally, more than 424 million hectares of topsoil (0–30 cm) and 833 million hectares of subsoil (30–100 cm) are impacted by salinization and alkalization. This trend is continuing to escalate. By 2024, the global total area of soil affected by alkalization and salinization is projected to reach 1.381 billion hectares, accounting for 10.7% of the total land area (FAO). In China, the extent of alkaline and saline-alkaline soil covers nearly 100 million hectares, with the Songnen Plain in Northeast China making up 3.73 million hectares of typical alkaline soil—this region, together with Victoria in Australia and California in the United States, constitutes three of the world’s most notable alkaline soil distribution areas [5]. As climate patterns shift due to global warming, decreased precipitation and increased evaporation exacerbate water scarcity, while poor agricultural practices (e.g., excessive fertilizer use, flood irrigation, and lack of rotational fallow) further accelerate soil alkalization [6]. Projections indicate that, in the near future, nearly 50% of global cultivated land may be affected by alkalization [7]. China’s staple food crops, including rice (*Oryza sativa* L.), maize (*Zea mays* L.), and wheat (*Triticum aestivum* L.), are glycophytes highly sensitive to alkaline stress [8]. Notably, roughly one-third of China’s alkaline land has reclamation potential; utilizing this reclaimed land for agriculture could play a crucial role in mitigating future food shortages [9]. Cultivating alkali-tolerant crops is therefore an effective strategy to alleviate the adverse effects of soil alkalization, making it essential to investigate key factors, germplasm resources, and molecular mechanisms underlying plant alkali tolerance to support high-quality agricultural development in China.

Soil salinization is commonly accompanied by alkalization, particularly in low-precipitation regions [10], but the two stressors differ fundamentally—alkaline stress poses more complex and severe threats to plants. Saline-alkali soil encompasses a spectrum of soils with varying salinization and alkalinity levels: soils are classified as saline when surface salt content exceeds 0.6% (dominated by neutral salts like NaCl and Na_2_SO_4_, with neutral pH), while alkaline soils have surface salt content below 0.5%, high levels of sodium carbonate (Na_2_CO_3_) and sodium bicarbonate (NaHCO_3_), and pH above 8.5 [11]. Alkaline stress severity is further categorized by salt content and pH: mild (salt content < 3 g/kg, pH 7.1–8.5), moderate (salt content 3–6 g/kg, pH 8.5–9.5), and severe (salt content > 6 g/kg, pH 9.5–10.5) [12]. Salt stress primarily causes osmotic stress (via reduced soil water potential) and ion toxicity (from Na^+^/Cl^−^ accumulation), while alkaline stress induces additional harm: elevated pH disrupts cellular pH homeostasis, damages cell membrane integrity, reduces root activity, impairs photosynthesis [13,14], and accelerates mineralization of soil organic matter (e.g., carbon, nitrogen, phosphorus, sulfur)—diminishing nutrient availability [15]. Numerous studies confirm that alkaline stress alone (even without salt stress) causes more severe nutrient imbalance, osmotic regulation failure, and antioxidant system impairment than salt stress, leading to stronger inhibition of plant growth [16,17,18,19].

Extensive research has revealed that plants have evolved diverse, coordinated mechanisms to cope with alkaline stress. A well-documented strategy is the synthesis and secretion of organic acids, which act as buffers, antioxidants, and ion balancers. For instance, tomato responds to alkaline stress by triggering compensatory biosynthesis of carboxylates (citrate, malate, succinate) in roots and leaves, using these anions to mitigate inorganic anion deficits [20]. *Puccinellia tenuiflora* primarily accumulates citric acid under alkaline stress, while Polygonaceae species rely on oxalic acid accumulation [21,22], highlighting species-specific differences in organic acid metabolism. Recent studies have expanded this understanding across more taxa: the alkali-tolerant forage *Leymus chinensis* upregulates genes encoding cytoplasmic malic enzyme (ME) and mitochondrial citrate synthase (CS) under alkaline stress, boosting root secretion of malate and citrate to lower rhizosphere pH by 1.2–1.5 units and activate soil available phosphorus [23]. In flax (*Linum usitatissimum*), pH 9.0 stress induces a 3–5 fold increase in citrate and tartrate levels, with secretion mediated by the *LuALMT1* transporter gene—expression of which correlates positively with alkali tolerance [24]. Beyond direct stress mitigation, organic acids also function as central intermediates in plant carbon metabolism: they act as signaling molecules, modulate transmembrane transport, facilitate protein modifications (e.g., acetylation, succinylation), and enhance soil nutrient absorption—all of which collectively improve plant alkali resistance [25,26,27,28,29].

In addition to secreting organic acids, plants employ multiple other strategies to cope with alkaline stress. For instance, they can enhance their resistance by increasing the synthesis of endogenous hormones such as abscisic acid (ABA), indole-3-acetic acid (IAA), and jasmonic acid (JA). Under alkaline conditions, elevated levels of endogenous or exogenous ABA can activate the antioxidant defense system, thereby reducing reactive oxygen species (ROS) accumulation and mitigating oxidative damage [30].

GsMSRB5a, a methionine sulfoxide reductase B5a from *Glycine soja*, interacts with the Ca^2+^/calmodulin-dependent kinase GsCBRLK. This interaction helps suppress ROS formation by modulating the expression of genes involved in ROS signaling, biosynthesis, and scavenging, thereby enhancing alkaline stress tolerance [31].

Under alkaline stress, genes involved in stress perception, signal transduction, and defense mechanisms are differentially regulated to modulate the expression of associated proteins. Transcriptome sequencing analysis in *Arabidopsis thaliana* under high-pH stress has demonstrated that transcription factors such as AP2/EREBP, WRKY, NAC, and MYB play critical roles in this process [32].

Collectively, plants mitigate alkaline stress through the coordinated action of multiple pathways, thereby sustaining normal growth and development.

In this study, we aimed to investigate whether maize roots secrete organic acids as a mechanism to counteract alkali stress. To achieve this, we subjected 369 maize inbred lines to alkaline treatment in order to identify varieties with varying degrees of alkali stress tolerance. From this screening process, we identified two alkali-resistant and two alkali-sensitive varieties. By integrating phenotypic analysis with transcriptomic and metabolomic data, we pinpointed two candidate genes, which were subsequently validated using RT-qPCR. Specifically, the *Zm00001eb396990* gene (encoding asparagine synthetase 4) may play a regulatory role in amino acid metabolism and participate in the production of acidic substances in roots. The *Zm00001eb370000* gene (encoding cytokinin dehydrogenase) is likely involved in regulating root development under alkaline stress conditions. These genes are candidate targets for future functional validation in alkali stress response.

## 2. Results

### 2.1. Different Phenotypes of Four Inbred Lines to Alkali Stress

Among 369 maize inbred lines screened for alkali tolerance, two alkali-resistant (W172 and 18) and two alkali-sensitive (Mo113 and fusheA) lines were selected for detailed analysis.

In water, all lines showed a similar pattern, with an initial pH decline that quickly stabilized (Figure 1A). Under alkaline stress, pH trends were comparable across lines, but resistant varieties adjusted more rapidly, reaching a stable pH around 6 h, whereas sensitive varieties stabilized later, around 12 h. Despite differences in stabilization timing, all lines ultimately reached similar pH levels (Figure 1B).

These findings indicate that maize roots respond to alkaline stress by actively modulating rhizosphere pH. The faster adjustment in resistant varieties suggests a more effective and rapid regulatory mechanism, potentially contributing to their enhanced tolerance.

To investigate maize responses to alkaline stress, four varieties were grown under controlled soil conditions for 14 days, with continuous alkaline treatment, while controls received regular watering.

Under alkaline stress, root responses differed between resistant and sensitive lines. Alkali-resistant varieties showed reduced root length compared to controls, but root morphology remained largely unchanged. In contrast, sensitive varieties maintained root length, but exhibited notable morphological changes, including fewer lateral roots (Figure 2A–E). Shoot growth was largely unaffected, except for fusheA, which showed a slight increase in root length after treatment (Figure 2F,G). Overall, resistant varieties experienced a more pronounced reduction in root length, while shoot length was less affected compared with sensitive varieties.

Analysis of fresh and dry biomass revealed significant reductions in all varieties after alkaline treatment, with sensitive lines showing greater loss than resistant ones (Figure 2H–O). Root biomass decreased more than shoot biomass across all varieties, highlighting roots as the primary site of stress impact. Among the lines, fusheA exhibited the highest weight loss, whereas variety 18 had the lowest. In resistant varieties, weight loss was mainly due to shortened roots, while in sensitive varieties it resulted from reduced lateral root number and altered root morphology.

These results suggest that alkali-resistant varieties maintain root morphology and potentially secrete organic acids to alleviate stress, whereas sensitive varieties experience structural root changes that likely impair organic acid secretion, contributing to increased susceptibility.

### 2.2. Metabolic Profiling of Alkali-Stressed Maize Varieties

To investigate the metabolic response of maize to alkali stress, metabolome profiling and principal component analysis were performed on root exudates of four maize varieties under water and alkali treatments. A clear separation was observed between treatments, with PC1 explaining 31% of the total variation (Figure 3A), indicating distinct metabolic responses to alkali stress.

Orthogonal partial least squares discriminant analysis (OPLS-DA) showed that approximately half of the metabolite features in each variety (46–51%) were associated with alkali stress (VIP ≥ 1). A total of 132–169 differentially expressed metabolites (DEMs) were identified across the four varieties, including amino acids, organic acids, lipids, and nucleotides (Appendix A). KEGG enrichment analysis revealed that these DEMs were mainly involved in alanine, aspartate, and glutamate metabolism; pyrimidine metabolism; and arginine and proline metabolism (Appendix A).

Venn analysis identified 62 DEMs shared among all varieties (Figure 3C). Notably, malonic acid, which showed stable levels under water treatment, accumulated significantly under alkali stress—particularly in resistant varieties (Figure 3D). Further pathway analysis indicated that this compound is mainly derived from pyrimidine metabolism, where intermediate metabolites decreased while terminal products, including malonic and methylmalonic acids, increased after alkali treatment (Figure 3E and Appendix A).

Together, these findings suggest that maize secretes malonic acid to adjust rhizosphere pH under alkali stress. Resistant varieties exhibit higher secretion capacity, which may underlie their greater tolerance to alkaline conditions.

### 2.3. Comparative Transcriptomic Analysis of Responses to Alkali Stress of Two Resistant Varieties and Two Sensitive Varieties

To elucidate the molecular basis of alkali tolerance, transcriptome sequencing was performed on roots of two alkali-resistant (W172, 18) and two alkali-sensitive (Mo113, fusheA) maize varieties under control and alkali treatments. Correlation analysis and PCA confirmed high reproducibility among biological replicates, with clear separation between treatment conditions (Figure 4A and Appendix A).

Comparative analysis identified substantial transcriptional reprogramming under alkali stress. Specifically, 2979 and 3066 differentially expressed genes (DEGs) were detected in the resistant varieties W172 and 18, while 4743 and 1909 DEGs were identified in the sensitive varieties Mo113 and fusheA, respectively (Figure 4B). In all genotypes, the number of up-regulated genes exceeded that of down-regulated genes, suggesting that alkali stress predominantly triggers gene activation. Heatmap analysis further revealed distinct expression profiles between resistant and sensitive varieties (Appendix A), implying divergent regulatory responses to alkali stress.

Collectively, these results indicate that transcriptional activation plays a key role in maize adaptation to alkali stress, with resistant and sensitive varieties exhibiting differential patterns of gene expression adjustment.

Considering the distinct responses of alkali-tolerant and sensitive varieties to alkaline treatment, we conducted KEGG enrichment analysis on the differentially expressed genes (DEGs) from the four maize alkaline treatment groups and the control group. The results revealed that these four varieties were significantly enriched in metabolic pathways and secondary metabolite biosynthesis pathways (Appendix A).

The enrichment of DEGs in these pathways may suggest that these pathways constitute one of the primary mechanisms by which maize adapts to alkaline stress. Furthermore, we observed that multiple hormone signaling pathways and the Mitogen-Activated Protein Kinase (MAPK) signaling pathway were significantly enriched in most samples (Appendix A), indicating that plant hormone signaling transduction and alkali stress signal transmission play a key role in the process of maize adaptation to alkali stress.

To rapidly and precisely identify the key genes responsible for the differences between alkali-tolerant and sensitive varieties, the Venn analysis method was employed. The upregulated differentially expressed genes in alkali-tolerant varieties were compared with the downregulated differentially expressed genes in sensitive varieties (Figure 4C). Additionally, the downregulated differentially expressed genes in alkali-tolerant varieties were compared with the upregulated differentially expressed genes in sensitive varieties (Figure 4D). Ultimately, a total of five candidate genes were identified. Among them, *Zm00001eb396990* and *Zm00001eb370000* are, respectively, involved in amino acid metabolism regulation and plant hormone signal transduction and may play significant roles in the process of plants responding to alkaline stress.

Notably, the candidate gene *Zm00001eb396990* encodes asparagine synthetase, which plays a key role in the alanine, aspartate and glutamate metabolic pathways, and is closely related to pyrimidine metabolism and β-alanine metabolism. Based on the above analysis, we believe that this gene plays an important role in regulating the resistance of maize to alkaline stress.

We randomly selected eight DEGs (*Zm00001eb031830*, *Zm00001eb367120*, *Zm00001eb072990*, *Zm00001eb400300*, *Zm00001eb393460*, *Zm00001eb312700*, *Zm00001eb017030*, *Zm00001eb038250*) from the transcriptome data and assessed their expression level changes using RT-qPCR. The results were in agreement with the trends observed in the transcriptome data, thereby validating the authenticity and reliability of the transcriptome data (Appendix A). Then we employed RT-qPCR to analyze the expression levels of the two candidate genes in four maize varieties under both alkaline and water treatment conditions. The results indicated that, compared with the water-treated control group, the expression of the *Zm00001eb396990* gene was significantly upregulated in the two alkaline-resistant varieties following alkaline treatment, whereas it was downregulated in the alkaline-sensitive varieties (Figure 4E).

This result is in agreement with our prior transcriptome analysis, indicating that the differential expression of this gene might serve as the primary factor contributing to the disparity in malonic acid accumulation between alkali-resistant and alkali-sensitive varieties. By contrast, the expression of the *Zm00001eb370000* gene was downregulated in alkali-resistant varieties while upregulated in alkali-sensitive varieties (Figure 4F).

### 2.4. Integrative Analysis of Transcriptome and Metabolome Data Revealing the Mechanism of Maize Response to Alkaline Stress

To elucidate the metabolic pathways underlying the differences in alkali stress tolerance among various maize varieties, we performed a transcriptome and metabolome conjoint analysis on samples from four maize varieties treated with either alkali or water.

The co-enrichment analysis of KEGG pathways for the transcriptome and metabolome revealed that the DEGs and DEMs of the four maize varieties after alkali treatment were mainly enriched in metabolic pathways and secondary metabolite biosynthesis pathways. This indicates that metabolic pathways and secondary metabolite biosynthesis pathways are crucial for maize to cope with alkali stress.

It is noteworthy that the metabolic pathways of alanine, aspartate, and glutamate were significantly enriched once more in transcriptome and metabolome KEGG pathways co-enrichment, suggesting their critical role in responding to alkaline stress (Appendix A). The key differentially expressed gene, *Zm00001eb396990*, within this pathway exhibited a significant correlation with differentially expressed metabolites across four maize groups. The analysis revealed that the *Zm00001eb396990* gene was strongly associated with malonic acid (pme1975) across the four maize inbred lines. A positive correlation was observed in the two alkali-resistant varieties, whereas a negative correlation was identified in the two alkali-sensitive varieties (Appendix A). These findings align with the upregulation of this gene in alkali-resistant varieties and its downregulation in alkali-sensitive varieties following alkaline treatment.

Based on the research results presented in this paper, we have developed a schematic diagram illustrating the secretion of organic acids by maize roots as a mechanism for resisting alkaline stress (Figure 5).

## 3. Discussion

Our study provides valuable insights into the molecular mechanisms underlying maize alkaline stress tolerance, particularly focusing on the role of organic acid secretion and associated gene regulation. The findings highlight the adaptive strategy of maize roots to alkaline stress by secreting organic acids to modulate soil pH, thereby mitigating the adverse effects of alkaline conditions.

In this study, we observed significant alterations in the morphological characteristics of both sensitive and resistant plant varieties under alkaline stress conditions (Figure 2A–D). These changes are likely an adaptive response to alkaline stress. Previous studies have found that under saline-alkali stress, roots first perceive the stress information and gradually transmit it to the above-ground part. The root surface area, the number of root tips, the leaf area and photosynthetic rate mainly explained the response of seedling biomass to saline-alkali stress [33]. Furthermore, research on cotton and *Leymus chinensis* has demonstrated that these plants can enhance their adaptability to saline-alkali stress through an increased root-shoot ratio and specific root length [34,35]. In addition to morphological adaptations to alkali stress, plants can also mitigate the effects of alkali stress through various physiological mechanisms. Saline-alkaline conditions elevate soil solute potential through sodium ion accumulation, depressing soil water potential below plant cellular thresholds. This hydraulic imbalance triggers cellular dehydration, inducing dual osmotic and physiological drought stress [36]. In response, plants accumulate compatible solutes (such as proline, betaine, polyamines, polyols, soluble proteins, and sugars), to counterbalance water potential deficits through osmoregulation. This phenomenon is evident in both sorghum, wheat and other plants [37,38,39,40].

The identification of two key genes, *Zm00001eb396990* (asparagine synthetase gene) and *Zm00001eb370000* (cytokinin dehydrogenase gene), underscores the complexity of the regulatory network governing organic acid production and plant responses to alkaline stress. Following alkaline treatment, differential expression of the key gene *Zm00001eb396990* was observed in the alanine, aspartate, and glutamate metabolic pathways in both alkaline-resistant and alkaline-sensitive varieties. This differential expression subsequently influenced the closely related pyrimidine metabolic pathway and β-alanine metabolic pathway. Both pathways are capable of metabolizing to produce malonic acid. Following alkali treatment, the levels of intermediate metabolites in the pyrimidine metabolic pathway were observed to decrease. We hypothesize that after alkali treatment, maize predominantly utilizes the pyrimidine metabolic pathway to produce malonic acid as a response to alkali stress. The disparity in the capacity to accumulate malonic acid between alkali-resistant and sensitive varieties contributes to the variation in tolerance to alkali stress.

While our findings strongly link the identified genes to the alkaline stress response, key mechanistic aspects remain unclear. For example, it is not yet understood how the asparagine synthetase gene (*Zm00001eb396990*) regulates flux through the pyrimidine and β-alanine pathways to promote malonic acid production. The enzymatic and regulatory steps connecting this gene to metabolic remodeling require further investigation. Additionally, the specific transporters responsible for secreting malonic acid and other organic acids into the rhizosphere have not been identified. Characterizing these transporters is essential to fully elucidate the pH modulation mechanism.

Previous studies on rice have shown that cytokinin inhibits the elongation and development of the main root but promotes the development of crown roots [41]. The differential expression of the cytokinin dehydrogenase gene (*Zm00001eb370000*) may explain contrasting root architecture responses to alkali stress. Resistant varieties exhibit significant primary root shortening with minimal change in lateral and crown roots, whereas sensitive varieties show a marked reduction in lateral and crown root number without significant alteration in primary root length. We plan to knockout or knockdown the *Zm00001eb370000* gene in resistant lines and overexpress it in sensitive lines. If this gene is central to the adaptive trait, altering its expression level is expected to modify the root architectural response to alkaline stress. Meanwhile, measuring tissue-specific cytokinin levels will help elucidate the underlying hormonal dynamics.

We propose that resistant varieties modulate root development via cytokinin to sustain organic acid secretion and pH modulation. In contrast, sensitive varieties may adopt a survival strategy by suppressing lateral root formation. A schematic model of this response is provided (Appendix A).

Integrated transcriptomic and metabolomic analyses uncovered a complex regulatory network, indicating distinct coping strategies. The correlation analysis of differentially expressed metabolites and candidate genes in W172 after alkali treatment showed that only a few metabolites were correlated with the expression of the asparagine synthase gene (Appendix A). We infer that as an alkali-tolerant maize variety, W172 may possess multiple mechanisms to cope with alkaline stress, among which the alanine, aspartate and glutamate metabolic pathway regulated by Zm00001eb396990 is just one of them. Although only a few metabolites related to Zm00001eb396990 were detected, the presence of malonic acid indicates that malonic acid secretion remains an effective alkali stress tolerance strategy in W172.

This study demonstrated that maize possesses the ability to modulate amino acid metabolism, which may subsequently influence root secretion of organic acids, thereby regulating the rhizosphere pH. Several studies have confirmed that organic acids can maintain pH stability within plant cells in response to alkaline stress [42,43]. In addition to the aforementioned pathways, plants display a diverse array of responses to alkali stress. For example, they maintain Na+-K+ ion homeostasis through channel proteins and transporters [44], enhance resistance to oxidative damage via antioxidant enzymes and antioxidants [45], and upregulate endogenous hormone synthesis by modulating relevant gene expression [46]. Despite these known mechanisms, many aspects of plant responses to alkali stress remain to be fully elucidated.

In conclusion, this study has significantly enhanced our understanding of maize responses to alkaline stress and provides a scientific foundation for developing strategies aimed at improving crop productivity in alkalinity-affected regions.

## 4. Materials and Methods

### 4.1. Plant Materials

The original seeds of the 369 maize inbred lines used for screening salt-alkali resistant and sensitive varieties were kindly provided by Professor Hongjun Liu from Shandong Agricultural University.

### 4.2. Plant Growth Condition

In April 2023, 369 maize inbred line materials were sown at the field base of Wenyangtian Modern Agricultural Industrial Park of Shandong Agricultural University, located in Nanqiu Village, Bianyuan Town, Feicheng City, Shandong Province (116°52′43″ E, 35°58′55″ N), for seed propagation purposes.

The seeds of each inbred line were screened before sowing to ensure that it was plump and approximately the same size. Prior to sowing, the seeds must undergo sterilization. Specifically, immerse the seeds in 70% ethanol and wash for one minute, followed by three rinses with deionized water. Then wash in 2% sodium hypochlorite solution for 10–15 min, shaking gently every 3 min. Finally, it is washed with deionized water 5–6 times and can be sown in nutritious soil. The seedlings were grown in an artificial climate chamber and cultured at 28 °C under long day conditions (16 h of light/8 h of darkness). Light intensity 50,000 Lux, air relative humidity 60–80% [47].

### 4.3. Preparation of Alkali Treatment Solution

To prepare an alkaline solution with a pH of 10.5: According to the required volume of the alkaline solution, measure an equal volume of deionized water and transfer it into a beaker. While continuously monitoring the pH, gradually add sodium carbonate anhydrous to the deionized water until the pH reaches 10.5. Measure an equal volume of untreated deionized water as a control. Seal the beaker with plastic wrap to prevent changes in the pH of the solution due to atmospheric carbon dioxide.

### 4.4. Screening of Maize Resistant Varieties and Sensitive Varieties

Previous studies have demonstrated that plants respond to alkaline stress by secreting organic acids to regulate rhizosphere pH [48]. Therefore, we employed the measurement of rhizosphere pH to screen for resistant varieties and sensitive varieties.

In our preliminary experiment, roots of 14-day-old seedlings from the maize B73 inbred line were hydroponically cultured in an alkaline solution for 24 h, with pH measurements taken at 3 h intervals. We observed that the most pronounced decline in pH occurred at approximately 6 h (Appendix A). Accordingly, 369 maize inbred lines were screened based on their pH values after 6 h of alkaline treatment (Appendix A).

### 4.5. Alkaline Treatment pH Curve Measurement

The roots of the seedlings cultured for 14 days were washed and placed in deionized water for 24 h to adapt to the hydroponic environment. Uniform seedlings were allocated to two treatment groups using a randomized block design, with each group comprising three biological replicates (*n* = 3 plants per replicate). The seedlings were put into a hydroponic box, 200 mL of alkali solution was added to the treatment group, and 200 mL of deionized water was added to the control group. The pH value of the solution was measured every 3 h, and the pH change curve was drawn for a total of eight measurements.

### 4.6. Alkali Treatment Phenotypic Evaluation

Seedlings for phenotypic identification must be from the same batch of the same year. The seeds were planted in nutrient soil and divided into two groups. The seeds were cultured in an artificial climate chamber and watered every two days. The same volume of alkali solution and deionized water was added to the treatment group and the control group, respectively. After two weeks of culture, the seedlings were washed for photography and phenotypic measurement. Shoot length and root length are measured with a measuring ruler. The fresh and dry weights of the plants are measured using Metle-tolidor precision balances (Mettler-Toledo, 1900 Polaris Parkway, Columbus, OH, USA). Dry the surface of the plant with absorbent paper before measuring fresh weight. After the fresh weight was measured, the plants were placed in the oven at 65 °C for 24 h to the weight remained constant, and then the dry weight of the plants was measured. In phenotypic statistics, 15 samples were measured in each group, *t*-test was used for shoot length, root length, fresh weights and dry weights statistical analysis. The weight loss rate was determined by subtracting the sample weight in the alkali treatment group from that in the control group, and then dividing the difference by the initial weight of the control group samples, *n* = 3. Two-way ANOVA was used for the statistical analyses.

### 4.7. Transcriptome and Metabolome Analysis

Previous experiments have determined that the pH drop rate of maize hydroponic solution is the highest at about 6 h after alkali treatment. Therefore, maize roots and hydroponic solution treated for 6 h were selected for the transcriptome and metabolome analysis detection by Metware Metabolic Biotechnology Co., LTD (Wuhan, China).

#### 4.7.1. Transcriptome Analysis

According to established protocols described in previous studies [49], transcriptome analysis was performed. RNA extraction, quality assessment, library preparation, and high-throughput sequencing were carried out by Metware Biotechnology Co., Ltd. (Wuhan, China) using the Illumina platform (Illumina, Inc., San Diego, CA, USA). Raw sequencing data were processed with fastp [50] to generate Clean Data under the following filtering conditions: paired-end reads were discarded if either read contained more than 10% N bases, or if more than 50% of the bases in any read had a quality score (Q) ≤ 20. All subsequent analyses were performed using the resulting high-quality clean reads.

After obtaining Clean Data, the reads were aligned to the *Zea mays* L. reference genome (Zm-B73-REFERENCE-NAM-5.0.fa.gz, https://download.maizegdb.org/Zm-B73-REFERENCE-NAM-5.0/, accessed on 15 October 2025) using HISAT2 (Version 2.2.1) [51]. Novel transcripts were then predicted with StringTie (Version 2.1.6) [48], and the corresponding novel genes were functionally annotated against multiple databases (KEGG, GO, NR, Swiss-Prot, TrEMBL, and KOG) using diamond (Version v2.0.9) [52] with an E-value cutoff of 1 × 10^−5^. Transcription factors were identified with the iTAK software (Version plant:1.7a) [53]. Gene expression quantification was performed using featureCounts (Version 2.0.3) [54], and FPKM values were calculated to represent expression levels. Differential expression analysis between the two groups was carried out with DESeq2 (Version 1.22.1) [55,56], applying the Benjamini–Hochberg method for *p*-value adjustment. Genes satisfying |log_2_FoldChange| ≥ 1 and FDR < 0.05 were considered differentially expressed. Finally, enrichment analyses of KEGG pathways and GO terms were conducted based on the hypergeometric test.

#### 4.7.2. Metabolome Analysis

The metabolomic profiling was performed following well-established protocols described in previous studies [57,58]. Maize hydroponic solutions from four varieties were thawed, vortexed for 30 s, and aliquoted into labeled 50 mL centrifuge tubes. After freezing at −80 °C overnight, samples were lyophilized under vacuum.

Lyophilized samples were reconstituted with 70% methanol containing an internal standard at a 30-fold concentration factor. The internal standard was prepared by dissolving 1 mg standard in 1 mL of 70% methanol to obtain a 1000 μg/mL stock solution, which was then diluted to 250 μg/mL with 70% methanol.

The mixtures were vortexed for 15 min, ultrasonicated in an ice-water bath for 10 min, and centrifuged at 12,000 r/min for 3 min at 4 °C. The supernatant was filtered through a 0.22 μm membrane into injection vials for LC-MS/MS analysis.

The liquid chromatography analysis was performed on an Agilent SB-C18 column (1.8 µm, 2.1 × 100 mm) with a mobile phase of 0.1% formic acid in ultrapure water (A) and 0.1% formic acid in acetonitrile (B). The gradient elution started at 5% B, increased linearly to 95% B over 9 min, held for 1 min, returned to 5% B from 10.0 to 11.1 min, and re-equilibrated until 14 min. The flow rate was 0.35 mL/min, the column temperature was 40 °C, and the injection volume was 2 μL.

Mass spectrometry used an ESI source at 500 °C with ion spray voltages of ±5500 V (positive/negative mode). Gas settings were GSI 50 psi, GSII 60 psi, and CUR 25 psi, with high collision-induced dissociation. MRM scans were performed using medium nitrogen collision gas, with individually optimized DP and CE for each transition, and specific MRM ion pairs were monitored according to metabolite elution.

Quantification was based on MRM mode, where the first quadrupole selected precursor ions, which were fragmented and filtered by the third quadrupole for specific product ions, enhancing accuracy and reproducibility. After data acquisition, peak areas were integrated and normalized for each metabolite across samples. After acquiring MS data from different samples, chromatographic peak areas were integrated for all detected metabolites. Peak integration for the same metabolite across different samples was subsequently corrected to ensure consistency [59].

### 4.8. RT-qPCR

The roots treated with alkali for 6 h were used for RT-qPCR. Total RNA from fresh maize roots was extracted using TransZol Up Plus RNA Kit (TransGen, Beijing, China). The cDNA was obtained by HiScript^®^II 1st Strand cDNA Synthesis Kit (+gDNA wiper) (Vazyme, Nanjing, China). RT-qPCR experiments were performed using PerfectStart^®^Green qPCR SuperMix (TransGen Biotech, Building 4, Zhongguancun Dongsheng International Science Park, No. 1 North Yongtaizhuang Road, Haidian District, Beijing, China) on a real-time PCR system LightCycler 96. The relative expression of the target gene was calculated by the 2^−(∆∆Ct)^ method.

### 4.9. KEGG Annotation and Enrichment Analysis

The identified metabolites were first annotated using the KEGG Compound database (http://www.kegg.jp/kegg/compound/, accessed on 15 October 2025) and subsequently mapped to the KEGG Pathway database (http://www.kegg.jp/kegg/pathway.html, accessed on 15 October 2025) for pathway analysis. We conducted KEGG pathway enrichment analysis to investigate the involvement of differentially expressed genes in specific biological pathways. The analysis was performed using the Advanced Enrichment Bubble Plot tool on the Metware Cloud platform (https://cloud.metware.cn, accessed on 15 October 2025). Analysis software package and version: Python 3.6.6; Function packages and versions: pandas 0.23.4. The analysis was conducted using default parameters [60].

### 4.10. Principal Component Analysis

Unsupervised PCA (principal component analysis) was performed by statistics function prcomp within R (www.r-project.org, accessed on 15 October 2025). The data was unit variance scaled before unsupervised PCA. The analysis was conducted using the Advanced PCA tool on the Metware Cloud platform (https://cloud.metware.cn, accessed on 15 October 2025). Analysis software package and version: R version 3.5.1; Function package and version: stats 3.5.1. The analysis was performed using default parameters [61].

### 4.11. Hierarchical Cluster Analysis and Pearson Correlation Coefficients

Hierarchical cluster analysis (HCA) and Pearson correlation coefficient (PCC) calculations for the samples and metabolites were performed using the R package ComplexHeatmap (version 2.12.0). The results are presented as a heatmap with dendrograms and a correlation heatmap, respectively. In the HCA, the normalized metabolite signal intensities (scaled to unit variance) are displayed using a color spectrum.

### 4.12. The Generation of Venn Diagrams, Line Graphs, and Column Graphs

Venn diagrams were performed using the Metware Cloud Advanced Venn Diagram Tool (https://cloud.metware.cn, accessed on 15 October 2025). The software package and version used for analysis were R version 3.5.1. The specific R package and version employed was VennDiagram 1.6.20. The fundamental analytical parameters applied were the default parameters. The line graphs and column graphs were generated using GraphPad Prism software Version 8.3.0 for Windows.

## 5. Conclusions

Our research reveals that under alkaline stress conditions, maize regulates the metabolic pathways of alanine, aspartic acid and glutamic acid by up-regulating the asparagine synthetase gene (*Zm00001eb396990*). This regulation may lead to the accumulation of malonic acid as a metabolic response to alleviate alkaline stress. Meanwhile, the down-regulation of the cytokinin dehydrogenase gene (*Zm00001eb370000*) may be related to the maintenance of plant root morphology, thereby enhancing the secretion of organic acids and improving its adaptive response to alkaline stress.

## Figures and Tables

**Figure 1 ijms-26-10632-f001:**
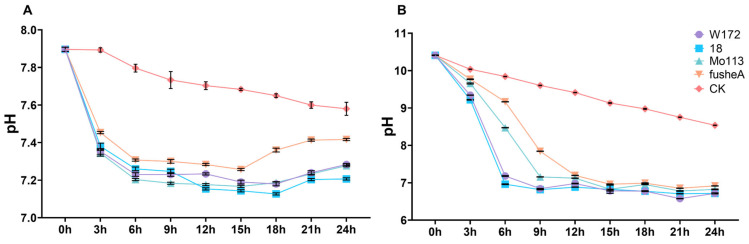
pH curve of different maize varieties treated with water and alkali for 24 h. (**A**) pH curve of water treatment for four maize inbred lines over time. (**B**) pH curve of alkali treatment for four maize inbred lines over time. CK, (Control Check). Curves of different colors represent different samples, purple for W172; blue for 18; green for Mo113; orange for fusheA; red for CK. Three repeated measurements were conducted at each time point.

**Figure 2 ijms-26-10632-f002:**
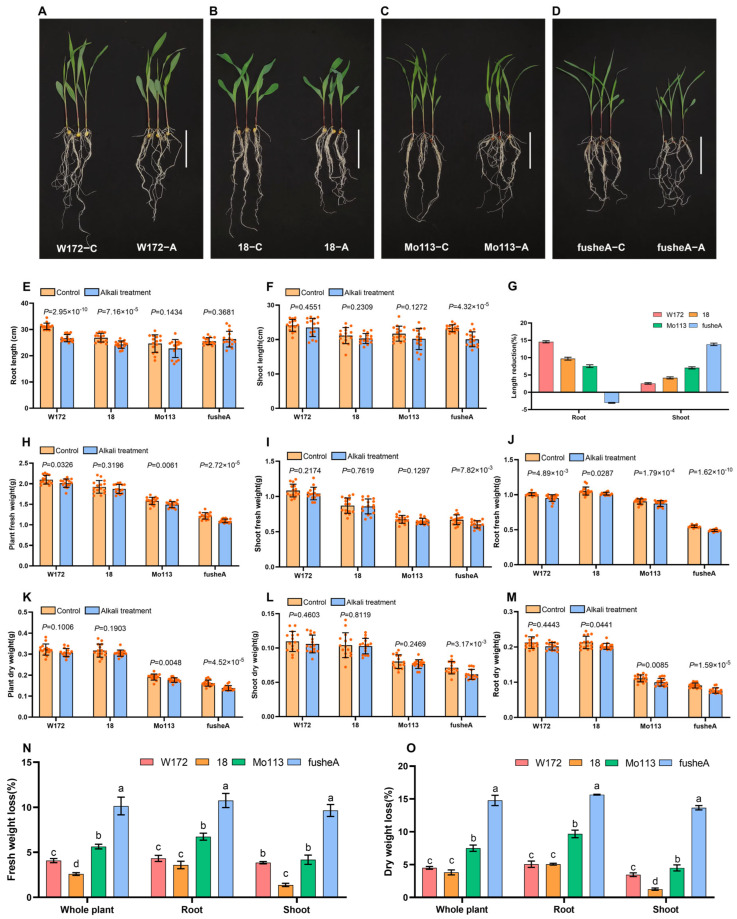
Observation and statistical analysis of phenotypic characteristics in four maize varieties after alkali treatment. Phenotypic characteristics of four maize inbred lines after alkali treatment. (**A**–**D**) Representative photographs after water treatment (**C**) and alkali treatment (**A**) [(**A**) W172, (**B**) 18 (**C**) Mo113 (**D**) fusheA]. Scale bar 10 cm. (**E**–**M**) Growth parameters with and without alkali treatment. (**E**) Root length (**F**) Shoot length (**G**) Root&Shoot length reduction ratio, Root length reduction ratio (W172, 14.56 ± 0.25%; 18, 9.70 ± 0.35%; Mo113, 7.55 ± 0.33%; fusheA −3.05 ± 0.09%) *n* = 3. Shoot length reduction ratio (W172, 2.57 ± 0.18%; 18, 4.16 ± 0.25%; Mo113, 7.06 ± 0.25%; fusheA 13.85 ± 0.28%) *n* = 3. (**H**) Plant fresh weight. (**I**) Shoot fresh weight. (**J**) Root fresh weight. (**K**) Plant dry weight. (**L**) Shoot dry weight. (**M**) Root dry weight. *T*-test between controls and alkali treatment, *p*-values are indicated above the columns. (**N**) Fresh weight loss rate. Whole plant (W172, 4.08 ± 0.19%; 18, 2.59 ± 0.12%; Mo113, 5.64 ± 0.21%; fusheA 10.15 ± 0.80%); root (W172, 4.33 ± 0.28%; 18, 3.59 ± 0.34%; Mo113, 6.74 ± 0.32%; fusheA 10.76 ± 0.64%); shoot (W172, 3.85 ± 0.10%; 18, 1.38 ± 0.15%; Mo113, 4.18 ± 0.42%; fusheA 9.65 ± 0.54%) *n* = 3 (**O**) Dry weight loss rat Whole plant (W172, 4.51 ± 0.17%; 18, 3.82 ± 0.31%; Mo113, 7.50 ± 0.39%; fusheA 14.78 ± 0.64%); root (W172, 5.06 ± 0.40%; 18, 5.07 ± 0.11%; Mo113, 9.68 ± 0.32%; fusheA 15.66 ± 0.05%); shoot (W172, 3.46 ± 0.23%; 18, 1.27 ± 0.14%; Mo113, 4.50 ± 0.37%; fusheA 13.66 ± 0.27%) *n* = 3 ANOVA, different letters indicate significant differences among the lines.

**Figure 3 ijms-26-10632-f003:**
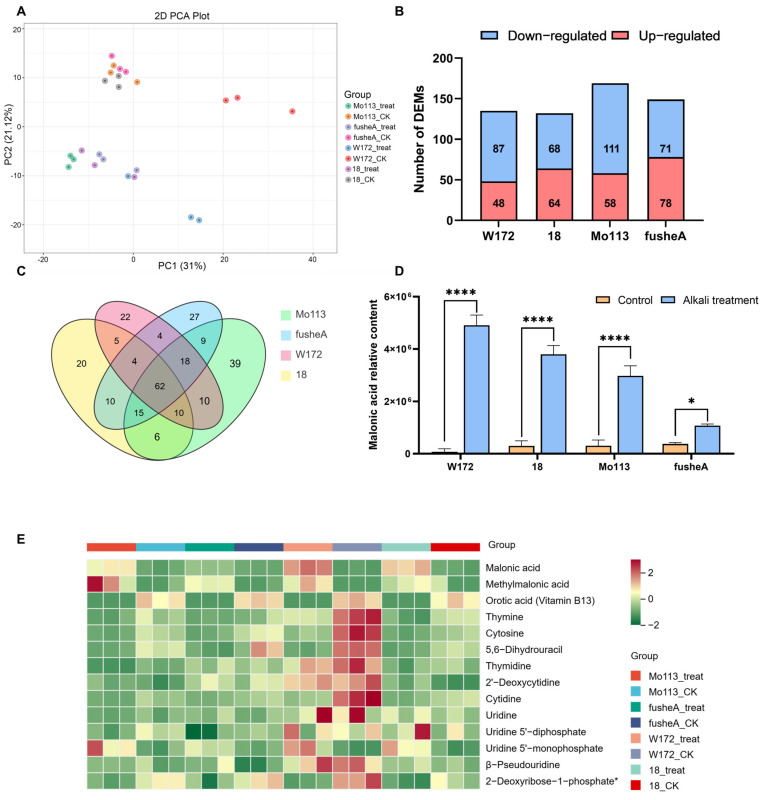
Metabolome analysis of four maize inbred lines after alkali treatment (**A**) PCA of metabolome samples. Four maize inbred lines were treated with deionized water (CK) and alkali treatment solution (treat) are indicated as different color dots, respectively. (**B**) The number of totals, upregulated, and downregulated DEMs in the different comparison groups. (**C**) Venn analysis of four maize inbred lines DEMs after alkali treatment. In the Venn diagram, each circle corresponds to a distinct comparison group. The numerical value within an overlapping region refers to the number of differential metabolites common to the corresponding groups, and the value in a non-overlapping region corresponds to the number of metabolites unique to that specific group. (**D**) The change in malonic acid content in the treatment solution of four maize varieties after alkali treatment. (Each sample contains three replicates, and the data are analyzed using a *t*-test. **** represent *p* < 0.0001, * represent *p* < 0.05) (**E**) Heat maps of differentially expressed metabolites in pyrimidine metabolic pathways following alkali treatment. The normalized signal intensities of the metabolites are visualized as a color spectrum, where different colors represent different relative levels (red indicates high levels, green indicates low levels). The metabolites marked with (*) indicate that there are isomers of these metabolites, and mass spectrometry technology cannot fully distinguish these isomers, so they are marked with an asterisk.

**Figure 4 ijms-26-10632-f004:**
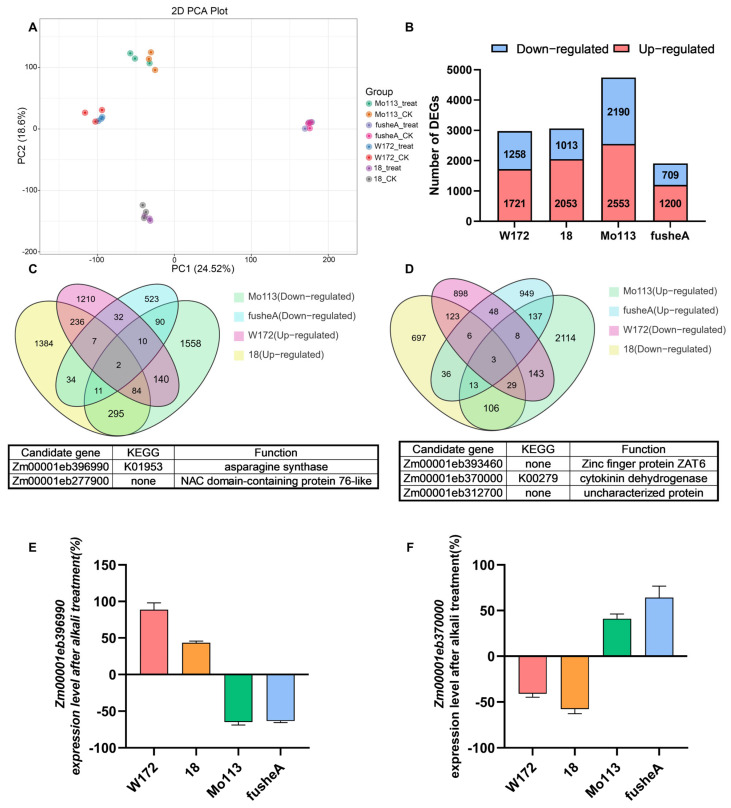
Transcriptome analysis of four maize inbred lines after alkali treatment. (**A**) PCA of transcriptome samples. Four maize inbred lines were treated with deionized water (CK) and alkali treatment solution (treat) are indicated as different color dots, respectively. (**B**) The number of total, upregulated, and downregulated DEGs in the different comparison groups. (**C**) Venn diagram of W172&18 upregulated DEGs and Mo113&fusheA downregulated DEGs. In the Venn diagram, each circle corresponds to a distinct comparison group. The numerical value within an overlapping region refers to the number of differential metabolites common to the corresponding groups, and the value in a non-overlapping region corresponds to the number of metabolites unique to that specific group. (**D**) Venn diagram of W172&18 downregulated DEGs and Mo113&fusheA upregulated DEGs. In the Venn diagram, each circle corresponds to a distinct comparison group. The numerical value within an overlapping region refers to the number of differential metabolites common to the corresponding groups, and the value in a non-overlapping region corresponds to the number of metabolites unique to that specific group. (**E**) Changes in expression level of *Zm00001eb396990* following alkali treatment. Each sample contains three replicates. (**F**) Changes in expression level of *Zm00001eb370000* following alkali treatment. Each sample contains three replicates.

**Figure 5 ijms-26-10632-f005:**
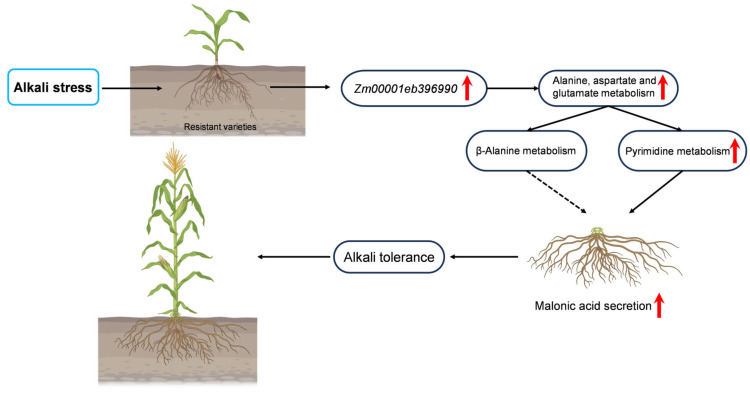
Schematic diagram of maize root organic acid secretion resisting alkali stress. Under alkaline stress conditions, upregulation of the Zm00001eb396990 gene in maize modulates alanine, aspartate, and glutamate metabolism along with pyrimidine metabolism. This regulatory effect further influences pyrimidine and β-alanine metabolic pathways, culminating in root exudation of malonic acid. The secretion of malonic acid contributes to rhizosphere pH homeostasis, conferring alkaline tolerance and thereby promoting enhanced growth and development in maize. The red arrow signifies an upward adjustment in expression levels.

## Data Availability

Data are contained within the article and Appendix A.

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
