# Peer review of "Integrated Transcriptome and Metabolome Analysis Identifies Key Genes Regulating Maize Tolerance to Alkaline Stress"

_ijms, 2025, doi:10.3390/ijms262110632_

Round 1

Reviewer 1 Report

Comments and Suggestions for Authors

The manuscript submitted by Liu et al. describes four maize lines that differ in their sensitivity to alkaline stress. Based on metabolome and transcriptome analyses, the authors identified malonic acid as a key root exudate that could mitigate alkaline stress. Furthermore, they identified two genes that may be related to the production of malonic acid and root system growth following exposure to alkaline stress.

The results are interesting, and the study appears solid and well executed. However, there are a number of points that could improve the manuscript.

  • Firstly, the title of the manuscript and the abstract do not align. While the title focuses on organic acids, the abstract focuses on the two genes that the authors identified as possibly important for resistance against alkaline stress.
  • The abstract's structure could be improved; it mixes results and conclusions. First describe what was done, then present the key findings in the same order as in the manuscript, and finally draw a conclusion. The adjectives 'resistant' and 'sensitive' should also be used in lines 25 and 26.
  • The introduction is interesting, but the main focus is on salt stress and worldwide problems with soil salinization. However, the manuscript only uses the stress factor of alkalinization. While it is clear that both are connected, the main focus should remain on alkaline soil. Additionally, I would appreciate more details on what is known about how plants respond to alkaline stress. This topic is addressed, but the information provided is unbalanced.
  • The methods section could be substantially extended to provide a clearer understanding of the work carried out.

a) Please provide a reference explaining why pH measurements are an effective way of measuring alkaline stress resistance.

b) Which previous results indicated to do the first measurements after 6 hours?

c) I could not fully follow how the alkaline solution was prepared. How stable is the pH of the solution? Is the solution buffered?

d) Is it correct that the water used in the first experiment had a pH higher than 7.5 already? What was added as a control? At what pH are maize plants usually grown?

e) If all plants can compensate for the high pH within nine hours and reach a stable pH value, why is the initial rapid decline so important? Is the difference between the lines observed between 3 and 9 hours significant, and how many independent replicates were analyzed?

f) Which tests were used for the statistical analyses (e.g., in Figure 2)? How was the weight loss rate calculated, and which statistical analysis was used? How many samples were analyzed?

g) Please submit the transcriptome data to a publicly available database.

h) What were the measurement parameters of the Sciex for the metabolome analysis? How accurate is the metabolite identification?

i) Please provide more details on the data analysis. Referring to a program available in the cloud is not sufficient. Which programs were used for the PCA, heatmap, etc.?

  • The results could be better structured and more concise. Please focus more on important details. Many of the exact numbers can be found in the figures, so they do not need to be repeated in the text.
  • I could not find the supplements. They were not uploaded to the system, and the link provided in the manuscript did not work, meaning there was no opportunity to review the supplementary figures and data.
  • Adding some paragraphs might make the text more readable.
  • Please try to be more concise. It is often mentioned at the beginning that something (e.g., a correlation) was found.
  • Please be careful not to discuss the results in the 'Results' section. For example, lines 345–351 could be moved to the 'Discussion' section. Similarly, lines 378–389 discuss the results. There are many more sentences that interpret and discuss the results rather than simply describing them.
  • Based on the finding that a cytochrome P450 was differentially expressed: Are there any differences in the levels of cytochrome among the lines and treatments?
  • The discussion section is rather generic and general. While an outlook is given on which additional experiments are needed to demonstrate that the scheme provided in Figure 8 is correct, I feel that a more in-depth discussion of open questions is lacking. How can the fact that only a few metabolites correlate with the asparagine synthase gene expression in W172_treat vs W172_CK be explained?
  • The figures and figure legends could be substantially improved. I suggest combining Figures 3 and 4, and moving Figures 3C–F and 4C to the supplements. Figure legends should be written so that they can be understood without reading the manuscript. Similarly, Figures 5 and 6 could be combined by moving Figures 5C–5F to the supplements. The panels in Figure 6 C and D could use the same color code as Figure 2 N and O.
  • Figure 7 is not very helpful. Which metabolites (or features) do all the numbers represent? Only show those that can be annotated to the compound class level, and indicate the class. Do the metabolites shown in Figure 4D also appear in Figure 7?

Author Response

Thank you very much for your careful review. We have made revisions based on the issues you raised, and the detailed content is presented in the Word document.

Reviewer 2 Report

Comments and Suggestions for Authors

The authors have provided a significant information about abiotic stress tolerance in crops and observe plant responses to high pH and also check the impact of metabolites involved in stress response which is a sound approach towards managing abiotic stress. The identification of malonic acid is acting as a buffer to regulate the mechanism of root architecture as well as in enhancing resistance towards stress. These finding will have wide spread implications towards practical application and will be key points in crop breeding and genetic engineering of major crops.

Author Response

Comments 1: The authors have provided a significant information about abiotic stress tolerance in crops and observe plant responses to high pH and also check the impact of metabolites involved in stress response which is a sound approach towards managing abiotic stress. The identification of malonic acid is acting as a buffer to regulate the mechanism of root architecture as well as in enhancing resistance towards stress. These finding will have wide spread implications towards practical application and will be key points in crop breeding and genetic engineering of major crops.

Response 1:

Thank you very much for your positive and encouraging comments on our manuscript. We are truly delighted that you found our work on abiotic stress tolerance significant and appreciated the approach of investigating the metabolic responses to high pH stress.

Your recognition of the potential role of malonic acid and the broader implications of our findings for crop breeding and genetic engineering is particularly gratifying. It is very encouraging to know that you consider these findings to have practical application potential.

Once again, we sincerely appreciate your time and valuable assessment of our work.

Reviewer 3 Report

Comments and Suggestions for Authors

Comments IN THIS current paper, maize roots were shown to secrete organic acid to regulate soil pH and then acclimate to an alkaline environment. Maize shown to modulate the metabolic pathways of alanine under alkaline stress conditions,  aspartic acid, and glutamic acid through the up-regulation of  the asparagine synthetase gene (Zm00001eb396990). This modulation results in the accumulation of malonic acid as a metabolic response to alleviate alkaline stress. Concurrently, the down-regulation of the cytokinin dehydrogenase gene (Zm00001eb370000) enables the plant to maintain root morphology and enhances the secretion of organic acids, thereby improving its adaptive response to alkaline stress. Two sensitivity lines and two resistance maize lines were select to perform transcriptome and metabolome conjoint analysis. two candidate genes: Zm00001eb396990 and Zm00001eb370000. The gene Zm00001eb396990 were identified as codes asparagine synthase, which plays a critical role in the metabolic pathways of alanine, aspartic acid, and glutamic acid. Please check carefully English language.  Collectively, the above results are good, and neat, figures including Correlation analysis of differentially expressed metabolites and candidate genes in four maize inbred line, some good figures and Venn diagrams show the analysis in good manner, The conclusions are very consistent with the evidence the topic is original and relevant to the field .

Author Response

Comments 1: Comments IN THIS current paper, maize roots were shown to secrete organic acid to regulate soil pH and then acclimate to an alkaline environment. Maize shown to modulate the metabolic pathways of alanine under alkaline stress conditions,  aspartic acid, and glutamic acid through the up-regulation of  the asparagine synthetase gene (Zm00001eb396990). This modulation results in the accumulation of malonic acid as a metabolic response to alleviate alkaline stress. Concurrently, the down-regulation of the cytokinin dehydrogenase gene (Zm00001eb370000) enables the plant to maintain root morphology and enhances the secretion of organic acids, thereby improving its adaptive response to alkaline stress. Two sensitivity lines and two resistance maize lines were select to perform transcriptome and metabolome conjoint analysis. two candidate genes: Zm00001eb396990 and Zm00001eb370000. The gene Zm00001eb396990 were identified as codes asparagine synthase, which plays a critical role in the metabolic pathways of alanine, aspartic acid, and glutamic acid. Please check carefully English language.  Collectively, the above results are good, and neat, figures including Correlation analysis of differentially expressed metabolites and candidate genes in four maize inbred line, some good figures and Venn diagrams show the analysis in good manner, The conclusions are very consistent with the evidence the topic is original and relevant to the field .

Response 1:

Thank you very much for your meticulous review of our manuscript and your valuable, constructive feedback. We greatly appreciate the time and effort you have dedicated to evaluating our work—your careful attention to both the scientific content and English expression has been incredibly helpful for improving the quality of our paper. 

Round 2

Reviewer 1 Report

Comments and Suggestions for Authors

Thanks to the authors for revising the manuscript and considering the reviewers' comments. There are a few additional aspects I recommend changing.

- lines 33-34 and lines 129-132: The results should be framed more carefully. The manuscript does not provide experimental evidence that the two genes play a regulatory role. The results only demonstrate a correlation, and it is not demonstrated that the two genes play a crucial role in alleviating alkali stress. However, this remains a valid hypothesis.

- Please improve all Figure legends and the font sizes of some graphs:

Figure 1 - Font size of x and y axis and the legend is too small. Maybe it could also be added "over time" (pH curve of water treatment for four maize inbred lines over time.)

Figure 2 - Important statistical information is missing and I recommend rephrasing some parts of the legend: Phenotypic characteristics of four maize inbred lines after alkali treatment. (A-D) Representative photographs after water treatment (C) and alkali treatment (A) [(A) W172, (B) 18.... Scale bar 10 cm. (E-M) Growth parameters with and without alkali treatment. ... (G) Root length reduction ratio.... Data shown are mean +- se (???) n=?. T-test between controls and alkali treatment, p-values are indicated above the columns.  (N)..... Data shown are mean +- se (???) n=?. ANOVA, different letters indicate significant differences among the lines.

Figure 3 - Please increase the font size in Panel A. Suggested heading: Metabolome analysis of four maize inbred lines after alkali treatment. For (D) please mention the number of replicates and the statistical test.

Figure 4 - Please increase the font size in Panel A. Suggested heading: Transcriptome analysis of four maize inbred lines after alkali treatment. For (E) please mention the number of replicates. Are the differences in expression statistically significant? (E and F) Changes in expression level of....

Figure 5: Schematic diagram (or alternatively scheme). Please briefly describe what is shown in the scheme.

Line 303: What is meant with once more?

Line 306: across the four maize inbred lines

- Lines 319-322: Please delete the sentence and the references

The Discussion could be better structured. I suggest to continue after line 326 with lines 373- 389.

As mentioned above, please be more careful with the wording; it has not been shown that the modulation of the amino acid biosynthesis enables the root system to secrete organic acids (line 390-392).

Similarly, please frame the conclusions more carefully!

Supplemental Figures:

- Differential accumulation of metabolites...

- For all Figures: please provide the number of replicates in all Figures (where data is aggregated).

Author Response

(The authors gave the same response as above.)
